# Studies Towards Hypoxia-Activated Prodrugs of PARP Inhibitors

**DOI:** 10.3390/molecules24081559

**Published:** 2019-04-19

**Authors:** Benjamin D. Dickson, Way Wua Wong, William R. Wilson, Michael P. Hay

**Affiliations:** 1Auckland Cancer Society Research Centre, School of Medical Sciences, Faculty of Medical and Health Sciences, University of Auckland, Private Bag 92019, Auckland 1010, New Zealand; w.wong@auckland.ac.nz (W.-W.W.); wr.wilson@auckland.ac.nz (W.R.W.); m.hay@auckland.ac.nz (M.P.H.); 2Maurice Wilkins Centre for Molecular Biodiscovery, University of Auckland, Symonds St, Auckland 1010, New Zealand

**Keywords:** PARP inhibitor, NU1025, hypoxia-activated prodrugs, hypoxia, prodrug, anti-cancer agents, tumor targeting, radiolytic reduction

## Abstract

Poly(ADP-ribose)polymerase (PARP) inhibitors (PARPi) have recently been approved for the treatment of breast and ovarian tumors with defects in homologous recombination repair (HRR). Although it has been demonstrated that PARPi also sensitize HRR competent tumors to cytotoxic chemotherapies or radiotherapy, normal cell toxicity has remained an obstacle to their use in this context. Hypoxia-activated prodrugs (HAPs) provide a means to limit exposure of normal cells to active drug, thus adding a layer of tumor selectivity. We have investigated potential HAPs of model PARPi in which we attach a bioreducible “trigger” to the amide nitrogen, thereby blocking key binding interactions. A representative example showed promise in abrogating PARPi enzymatic activity in a biochemical assay, with a ca. 160-fold higher potency of benzyl phthalazinone **4** than the corresponding model HAP **5**, but these *N*-alkylated compounds did not release the PARPi upon one-electron reduction by radiolysis. Therefore, we extended our investigation to include NU1025, a PARPi that contains a phenol distal to the core binding motif. The resulting 2-nitroimidazolyl ether provided modest abrogation of PARPi activity with a ca. seven-fold decrease in potency, but released the PARPi efficiently upon reduction. This investigation of potential prodrug approaches for PARPi has identified a useful prodrug strategy for future exploration.

## 1. Introduction

Poly(ADP-ribose)polymerases (PARP) are a family of enzymes involved in the synthesis of poly(ADP-ribose) (PAR) chains from NAD^+^. Of the eighteen described members, three (PARP-1, -2, -3) have defined roles in DNA damage repair [1,2,3,4]. The role of PARP-1 is the most clearly defined and it contributes most significantly to PAR synthesis. In response to single strand breaks (SSBs) in DNA, PARP-1 binds to DNA and attaches PAR chains to nuclear proteins (PARylation) including itself (autoPARylation). The PAR chains recruit base excision repair (BER) enzymes to the SSB and ultimately lead to dissociation of PARP [1,2,3,4,5]. This role in DNA damage repair lead to an early proposal that PARP inhibitors (PARPi) could find utility in cancer therapy [2].

PARP inhibitor cytotoxicity derives from a variety of mechanisms which have been well described elsewhere [6,7,8,9,10,11]. Briefly, catalytic inhibition of PARP stalls the BER process resulting in downstream DNA lesions when the replications forks collide with unrepaired SSB [7,12,13]. DNA lesions are also generated through the inability of inhibited PARP to dissociate from DNA, so-called “PARP trapping” [9,10,11]. In normal tissues, these lesions are repaired with high fidelity by homologous recombination repair (HRR), however, in HRR incompetent cell (e.g., *BRCA1* and *BRCA2* mutants) repair occurs via low fidelity pathways, including non-homologous end joining (NHEJ), resulting in an accumulation of errors and ultimately cell death [12,13,14]. This combination of a genetic defect and a pharmacological treatment combining to cause cell death is a form of “synthetic lethality” and has provided the context for clinical PARPi approvals to date [14,15,16].

In tandem with development of potent small molecule PARPi, increased investigation of PARP biology has established involvement of the PARP family in the wider DNA damage response [3,4]. In addition to involvement in BER, PARPs participate in HRR, canonical NHEJ (cNHEJ) and alternate end joining (alt-EJ), and have numerous interactions with nuclear proteins of unknown consequence [3,4,17,18]. Due to this widespread involvement, PARPi can sensitize cells to a variety of DNA damaging agents, and therefore combination with cytotoxic chemotherapies or radiotherapy has been proposed as an approach for treatment of HRR competent tumors [19,20]. However, studies have shown that use of PARPi in combination therapies often lead to normal tissue toxicity requiring reduction in the dose of either the PARPi or chemotherapeutic agent [21,22,23,24,25,26,27,28].

Hypoxia is a well-established feature of many solid tumors which contributes to both tumor progression and resistance to therapy [29,30,31,32,33,34]. As tumors grow, an oxygen gradient develops as its metabolic consumption outstrips the oxygen supply. Tumor vasculature lacks the organization of normal tissue vasculature which leads to tumor hypoxia, with chronic hypoxia due to oxygen diffusion limitations, and acute hypoxia caused by transient blockages or flow reversals [29,34].

We, and others, have demonstrated that hypoxia can be exploited to activate a prodrug selectively within a tumor [29,32,35]. These hypoxia-activated prodrugs (HAPs) rely on the different metabolic fates of a bioreducible functional group (i.e., a trigger) in oxygenated versus hypoxic environments. One such trigger, the nitroaromatic group, is reduced by one-electron reductases to a nitro radical anion [29,32]. Under normoxia, this radical anion is oxidized back to the parent nitro group, whereas under hypoxia, direct fragmentation of the radical anion, or further reduction to electron-donating hydroxylamino or amino groups leads to activated species [36]. This shift in electron density can activate the drug via fragmentation of a frangible linker (e.g., evofosfamide) [37] or through activation of a reactive centre (e.g., PR-104) [38].

We considered that tumor-selective delivery of a PARPi via a HAP would increase the therapeutic index of PARPi in combination with radiotherapy or chemotherapy. To explore this proposition we started with olaparib (Lynparza) **1** as an ideal “effector” for use in a HAP as it has nanomolar potency as a PARP-1 inhibitor and recently gained first-in-class registration in an BRCA mutant advanced ovarian cancer setting as a monotherapy [15,39].

The PARPi binding mode exemplified by olaparib **1** relies on a tridentate hydrogen-bond network which mimics the natural substrate nicotinamide, Figure 1. The phthalazinone carbonyl interacts with both Ser904-**O**H and Gly863-N**H** and the amide proton interacts with Gly863-C**O**. Additional interactions are formed by Tyr907 and Tyr986 forming π-stacking arrangements with bound inhibitor [40].

We predicted that the addition of a 2-nitroimidazolyl trigger to the phthalazinone NH of olaparib **1** would disrupt the bonding interaction with Gly863-**CO**, resulting in a detrimental effect on PARP inhibition. This concept has precedence in the work of Threadgill and co-workers who installed nitroheterocyclic triggers on a series of isoquinolin-1-ones **2**, Figure 2, and demonstrated modest abrogation of PARP inhibition [42,43]. Fragmentation of 2-nitrofuryl prodrugs **3a,b** and 2-nitroimidazolyl prodrug **3c** released effectors **2a–c**, respectively, following chemical reduction (NaBH_4_, Pd/C; SnCl_2_; Zn/NH_4_Cl) [42,43].

To build on this initial observation and to explore the potential of this prodrug approach we prepared a series of model compounds and related 2-nitroimidazolyl derivatives based on a series of PARPi. We prepared phthalazinone **4** as a representative of the structural core of olaparib **1** and the corresponding 2-nitroimidazolyl derivative **5** as a model HAP in order to assess disruption of PARP inhibition. To assess trigger fragmentation, we prepared derivatives (**3c**, **7**, **8**, **10**, **11** and **13**, Figure 3) of model PARPi (**2a**, **2c**, **6** and **9**, Figure 3) that represent the core of literature PARPi. We also considered other possible sites of the PARPi core for placement of a trigger. The PARPi NU1025 **12**, Figure 3, contains a phenol distal to the core binding motif common to PARPi. This phenol may interact with Glu988 via a hydrogen-bond bridge with water molecules (depicted as red spheres in Figure 1) [40]. We proposed that forming a 2-nitroimidazolyl methyl ether at this phenol would hinder this interaction and could undergo fragmentation following reduction, providing an alternate prodrug strategy [44,45].

We evaluated the potency of model HAPs **5** and **13** and the associated PARPi **4** and **12** in a biochemical assay (Reaction Biology Corp, Malvern, PA) as representatives of these two approaches and investigated the stability of all model HAPs in a radiolytic reduction assay.

## 2. Results and Discussion

### 2.1. Synthesis

#### 2.1.1. Synthesis of PARPi

Benzyl phthalazinone **4** was prepared from isobenzofuranone **14** via addition of hydrazine hydrate, Scheme 1 [39]. Benzamide **6** was prepared by alkylation of hydroxybenzoic acid **15** in the presence of potassium iodide and potassium carbonate [46]. The remaining amide PARPi scaffolds (**2a**, **2c,** and **9**, Figure 3) were sourced commercially. The phenol containing PARPi NU1025 **12** was prepared in four steps from benzoic acid **16** as previously described by Griffin et al. [47].

#### 2.1.2. Synthesis of Model HAPs

The key nitroimidazole alcohol intermediate **18** was prepared following the previously described route [35,48]. Conversion to chloride **19** was achieved by mesylation of **18** and in situ chloride displacement, Scheme 2 [44]. Alcohol **18** was converted to amine **21** via reductive amination of aldehyde **20**, prepared by oxidation of **18**. Aldehyde **20** was converted to secondary alcohol **22** by addition of a methyl titanium species as described by Winn et al. [49]. Oxidation to ketone **23** was followed by reductive amination to provide secondary amine **24**. Both amines **21** and **24** were used without isolation in subsequent reactions, due to their instability.

Preparation of model prodrugs **3c**, **5**, **10**, **11,** and **13** was achieved by combination of PARPi **2c**, **4**, **9**, **2a**, **12,** and **19** in the presence of a suitable base Scheme 3. LiHMDS furnished isoquinolin-1-one prodrugs **3c** and **11,** and sodium hydride was used for phthalazinone prodrugs **5** and **10**. Formation of the *N*-alkyl products was confirmed by comparison of the NCH_2_
^13^C NMR data to literature values. In O-alkyl products the OCH_2_ resonance appears between δ 50–65, whereas the NCH_2_ resonance in compounds **3c**, **5**, **10,** and **11** occurs in the range δ 30–50 as characteristic of *N*-alkylated compounds [50]. Phenoxy prodrug **13** was prepared using K_2_CO_3_ to deliver selective phenol alkylation.

In contrast alkylation of primary amide **6** did not proceed despite screening a number of bases. In an alternative approach phenol **25** was first converted to acid **26** and then the corresponding acyl chloride which was used directly to acylate amine **21** giving prodrug **7**, Scheme 4.

The fragmentation rates of 2-nitroheteroaromatic triggers are influenced by the ability of α-methylene substituents to stabilize developing positive charge in the fragmentation reaction [45]. Although an α-disubstituted trigger might be expected to provide maximal fragmentation rates these are extremely difficult to prepare [45,49]. We elected to prepare the α-substituted amide **8** to explore the influence of substitution on fragmentation rate. Formation of an acyl chloride from **26** and acylation of amine **24** yielded prodrug **8**, Scheme 4.

### 2.2. Biochemical PARP-1 Inhibition

The PARP-1 inhibitory activity of the compounds was determined for the PARPi/model HAP pairs **4**/**5** and **12**/**13** in a radiometric PARP-1 inhibition assay, Table 1. Both compounds **4** and **12** were potent PARPi. Model HAP **5** provided significant (ca. 160-fold) deactivation of PARP inhibition, consistent with disruption of the key hydrogen-bonding network between the amide of PARPi **4** and PARP-1, Figure 1. In contrast, phenol prodrug **13** only showed ca. seven-fold disruption of PARP-1 inhibition, consistent with the added 2-nitroimidazolyl ether interfering with a secondary interaction, such as with Glu988, but not disrupting the key binding interactions. 

### 2.3. Radiolytic Reduction

The ability of the model HAPs to release the effectors was assessed in a radiolytic reduction assay that provides obligate one-electron reduction with well-defined stoichiometry. Radiolysis of water generates both reducing (e^-^_(aq)_) and oxidizing (OH^•^, H^•^) radicals but the latter can be scavenged by formate ions to generate the reducing CO_2_^•-^ radical to give a total of 0.62 μmol.J^−1^ reducing radicals [51,52]. Compounds (10 μM) in anoxic 100 mM sodium formate/5 mM sodium phosphate buffer, pH 7.0, were irradiated (40 Gy) using a cobalt-60 source and analyzed by HPLC with in-line photodiode array absorbance and single-stage quadrupole mass spectrometry immediately after irradiation and compared with unirradiated controls. Samples were also analyzed following incubation for 5 h at 37 °C after irradiation to allow for slow fragmentation of hydroxylamine or amine intermediates to be observed. However, incubation did not alter the quantitation or identity of the species produced. Therefore, only immediate analysis results are reported. Typical results are illustrated for compounds **5**, Figure 4; **3c**, Figure 5; and **13**, with a summary of prodrug loss and product formation in Table 2. 

Radiolytic reduction of **5** resulted in an 85.3% decrease in diode-array signal, but we did not detect any released PARPi **4** in either the diode-array chromatogram (with comparison to authentic **4**) or the extracted ion chromatogram at the 237.1 m/z base peak, Figure 4, Table 2. Inspection of the mass spectrum revealed the hydroxylamine derivative of **5** (362.2 m/z) in the irradiated solution eluting at an earlier retention time, Figure 4. Previous studies have shown facile fragmentation of nitroheteroaromatic ethers after reduction to a hydroxylamine intermediate [45,53,54,55]. In this instance the stability of the hydroxylamine indicates that this system is too stable to release PARPi **4 [36,56]**.

We expanded our study to include analogous prodrugs based on the core scaffolds for other reported PARPi to see if the observed stability of the phthalazinone *N*-nitroimidazole framework was general [40]. We included bromoisoquinolinone **3c** as it has previously been shown to fragment in a chemical reduction system [43]. We assessed all the model prodrugs in our radiolytic reduction assay, Table 2. Irradiation of amide model HAPs **3c**, **7**, **8**, **10,** and **11** resulted in loss of prodrug that was broadly consistent with the expected four-electron stoichiometry. The major products were identified as the corresponding hydroxylamines, Table 2. No evidence for the release of effectors was detected, although authentic standards were detected with high sensitivity. Solutions of the prodrugs incubated for 5 h at 37 °C post irradiation provided no evidence of effector release, suggesting slow fragmentation is unlikely. This suggests that directly-linked model HAPs of the amides, including bromoisoquinolinone **3c**, Figure 5, do not readily fragment on reduction, in contrast to previous reports [42,43]. 

This discrepancy may be caused by the choice of reduction system. The formate radiolytic reduction system produces 4-electron reduction to the hydroxylamine species which we expected to fragment. However, we did not see evidence of 6-electron reduction to the more strongly electron-donating amino species. To address this we carried out reduction of **3c** with Zn/NH_4_Cl and assessed the resulting solution by LC/MS. After one hour exposure to Zn/NH_4_Cl in acetonitrile the solution was filtered and analysed by LC/MS. Comparison to a control solution allowed estimated loss of prodrug at 45% based on the diode-array signal. At the extracted *m/z* 333.0 signal we detected the corresponding amino species, Figure 6A, confirmed by its mass spectrum, Figure 6B. Re-analysis of the same sample after standing for 3 h showed no change in the ion count for the amine (data not shown) and, importantly, the expected fragmentation product **2c** was not detected despite ready detection of the authentic compound, Figure 6C,D. This result strengthens our observation that this type of prodrug does not release the desired PARP inhibitor, even when reduced to the corresponding amine. We are unable to account for observation of fragmentation in previous reports.

This disappointing result led us to explore a new prodrug approach. We prepared a 2-nitroimidazolyl ether linked to the phenol of NU1025 (**12**). Related prodrugs based on nitroheterocyclic ethers have been demonstrated to release effectors efficiently [44,45,49]. Reduction of the PARPi prodrug **13** did result in release of effector **12**, Table 2. Formation of **12** was detected in the diode-array chromatogram, Figure 7A, and confirmed by analysis of the mass spectrum in negative ionisation mode (**12** [M-H]^-^ 175.1 *m/z*; **13** M-H 314.0 *m/z*, Figure 7C,D) with an efficient yield of **12** at ca. 55-60% reduction of prodrug, Table 2.

In conclusion, the *N*-alkylated nitroimidazolyl prodrug system did provide deactivation of the PARPi as intended. However, this model prodrug system does not fragment upon reduction to the corresponding hydroxylamine or amine. In contrast, the phenol prodrug **13** shows only ca. seven-fold reduction in PARP inhibition. Importantly, the 2-nitroimidazolyl ether does fragment upon reduction and this provides a lead towards identification of novel prodrugs which can combine the efficient fragmentation of the ether linker with a larger deactivation of the PARPi.

## 3. Materials and Methods

### 3.1. Synthesis

#### 3.1.1. General

All non-aqueous reactions were carried out under a dry nitrogen atmosphere unless otherwise noted. DMF, DCM, and THF were purchased pre-dried and stored over molecular sieves from Acros Organics. All commercial reagents were used without purification. Flash column chromatography was carried out on a silica gel solid phase (Merck 230–400 mesh). Thin layer chromatography was carried out using Merck 60 F_254_ aluminium plates pre-coated with silica. Compounds were identified using UV fluorescence and/or staining with either vanillin in ethanolic sulphuric acid (with heating), 3, 5-dinotrophenylhydrazine in ethanolic sulfuric acid (with heating), ninhydrin in ethanol/glacial acetic acid (95:5) (with heating), or iodine on silica gel. Melting points were determined on an Electrothermal 2300 melting point apparatus. High resolution mass spectra (HRMS) were measured on an Agilent Technologies 6530 Accurate-Mass Quadrupole Time-of-Flight (Q-TOF) LC/MS interfaced with an Agilent Jet Stream electrospray ionization (ESI) source allowing positive or negative ions detection. Low resolution mass spectra (LRMS) were measured on a Surveyor MSQ mass spectrometer using an atmospheric pressure chemical ionization (APCI) mode with a corona voltage of 50 V and a source temperature of 400 °C. NMR spectra data were recorded on a Bruker Avance 400 spectrometer (400 MHz, ^1^H nuclei, 100 MHz, ^13^C nuclei). All chemical shift (δ) values were reported in parts per million (ppm) relative to tetramethylsilane (0.0 ppm) as an internal reference, coupling constants were reported in Hertz (Hz). Final products were analyzed by reverse-phase HPLC (Altima C18 5 μm column, 150 mm × 3.2 mm; Alltech Associated, Inc., Deerfield, IL) using an Agilent HP1100 equipped with a photodiode array detector. Mobile phases were gradients of 80% CH_3_CN/20% H_2_O (v/v) in 45 mM ammonium formate at pH 3.5 and 0.5 mL/min. Purity was determined by monitoring at 330 ± 50 nm.

#### 3.1.2. Procedures

**5-Bromo-2-((1-methyl-2-nitro-1*H*-imidazol-5-yl)methyl)isoquinolin-1(2*H*)-one (3c).** To bromoisoquinolinone **2c** (0.1 g, 0.69 mmol) in DMF (2 mL) was added a 1.0 M THF solution of LiHMDS (0.6 mL, 0.6 mmol) and the mixture was stirred for 2 h. Nitroimidazole **19** (0.08 g, 0.45 mmol) in DMF (2 mL) was added dropwise and the reaction was stirred for 2 h. NaI (0.002 g, 0.01 mmol) was added and the mixture was stirred overnight. A second addition of **19** (0.08 g, 0.45 mmol) was made and the mixture stirred a further 24 h. Water (2 mL) was added and the resulting mixture extracted with EtOAc (2 × 5 mL). The organic fractions were washed with water (5 mL), brine (5 mL) and dried over Na_2_SO_4_. Solvent was removed in vacuo and the crude product was purified by column chromatography (2:1, DCM, EtOAc) to yield **3c** (0.08 g, 49%) as an orange solid, mp 204–207 °C (lit. [43] mp 208–210 °C). δ_H_ (CDCl_3_) 8.41 (1H, dt, *J* = 7.8, 0.9 Hz, H-6), 7.94 (1H, dd, *J* = 7.7, 1.2 Hz, H-8), 7.39 (1H, t, *J* = 7.9 Hz, H-7), 7.23 (1H, s, H-4′), 7.14 (1H, d, *J* = 7.7 Hz, H-3), 6.94 (1H, dd, *J* = 7.7, 0.6 Hz, H-4), 5.30 (2H, s, CH_2_), 4.06 (3H, s, CH_3_). δ_C_ ((CD_3_)_2_SO) 160.3 (C = O), 145.8 (C), 136.4 (CH), 135.8 (C), 134.2 (C), 133.9 (CH), 128.0 (CH), 127.9 (CH), 127.3 (CH), 126.9 (CH), 120.0 (C), 104.0 (CH), 41.3 (CH_2_), 34.3 (CH_3_). HRMS calcd for C_14_H_12_^79^BrN_4_O_3_ (M + H) *m/z* 363.0093, found 363.0087; for C_14_H_12_^81^BrN_4_O_3_ (M + H) *m/z* 365.0073, found 365.0071. LRMS *m/z* 363.0 (100%, M^79^ + H), 365.0 (95%, M^81^ + H). HPLC purity: 95.5% (effector **2c**: not detected).

**4-Benzylphthalazin-1(2*H*)-one (4).** To a stirred suspension of benzalphthalide **14** (10 g, 45 mmol) in water (60 mL) was added aqueous NaOH (13 M, 15 mL) and the mixture was heated to 70 °C. Hydrazine hydrate (31 mL, 630 mmol) was added and the mixture stirred for 18 h. The reaction was cooled to room temperature and acidified with 8 M aqueous HCl to pH 4. After 10 min the resulting suspension was filtered and washed with water (2 × 50 mL) and diethyl ether (3 × 50 mL) to yield **4** (8.0 g, 75%) as a white solid, mp 198–200 °C (lit. [57] mp 200–201 °C). δ_H_ (CDCl_3_) 9.95 (1H, br s, NH), 8.47–8.43 (1H, m, H-8), 7.79–7.71 (3H, m, H-5, H-6, H-7), 7.33–7.00 (5H, m, Ar-H), 4.30 (2H, s, CH_2_Ph). δ_C_ (CDCl_3_) 161.1 (C = O), 146.5 (C), 137.8 (C), 133.6 (CH), 131.5 (CH), 130.0 (C), 128.9 (2 × CH), 128.7 (2 × CH), 128.5 (C), 127.2 (CH), 127.0 (CH), 125.6 (CH), 39.1 (CH_2_). LRMS *m/z* 237.2 (100%, M + H). HPLC purity: 99.2%. These data are in good agreement with literature values [57].

**4-Benzyl-2-((1-methyl-2-nitro-1*H*-imidazol-5-yl)methyl)phthalazin-1(2*H*)-one (5).** To phthalazinone **4** (0.33 g, 1.40 mmol) in DMF (2 mL) was added NaH (60% dispersion in mineral oil) (0.084 g, 2.10 mmol) and the resulting solution stirred for 30 min at room temperature. Nitroimidazole **19** (0.34 g, 1.94 mmol) was added and the solution was stirred for 30 min, then quenched on ice. The mixture was extracted with EtOAc (10 mL), washed with water (3 × 10 mL), dried over MgSO_4_ and solvent removed in vacuo. The crude product which was purified by column chromatography (1:1 X4, EtOAc) to yield **5** (0.10 g, 50%) as a yellow solid, mp 180–182 °C. δ_H_ ((CD_3_)_2_SO) 8.31 (1H, dd, *J* = 7.6, 1.4 Hz, H-8), 8.00 (1H, dd, *J* = 8.2, 1.0 Hz, H-5), 7.91–7.83 (2H, m, H-6, H-7), 7.30–7.24 (4H, m, Ar-H), 7.21–7.17 (2H, m, H-4′, Ar-H), 5.45 (2H, s, CH_2_N), 4.31 (2H, s, CH_2_Ph), 3.93 (3H, s CH_3_). δ_C_ ((CD_3_)_2_SO) 158.0 (C = O), 145.9 (C), 145.6 (C), 137.7 (C), 134.0 (C), 133.6 (CH), 132.0 (CH), 128.7 (2× CH), 128.4 (2× CH), 128.36 (CH), 128.36 (CH), 127.3 (C), 126.5 (2× CH), 125.8 (CH), 43.4 (CH_2_), 37.5 (CH_2_) 34.4 (CH_3_). HRMS calcd for C_20_H_18_N_5_O_3_ (M + H) *m/z* 376.1404, found 376.1412. LRMS *m/z* 376.2 (100%, M + H). HPLC purity: 96.1% (effector **4**: 0.3%).

**2-(Benzyloxy)benzamide (6).** To amide **15** (0.80 g, 5.83 mmol), K_2_CO_3_ (0.97 g, 7.02 mmol) and KI (1.17 g, 7.02 mmol) in acetone (30 mL) was added benzyl chloride (0.81 mL, 7.02 mmol), and the reaction stirred at reflux for 18 h. The solution was cooled and filtered and solvent removed in vacuo. The crude product was recrystallised from acetone to yield **6** (0.28 g, 21%) as a cream solid, mp 112–114 °C (lit. [58] mp 115–116 °C). δ_H_ (CDCl_3_) 8.25 (1H, dd, *J* = 7.8, 1.8 Hz, H-6), 7.73 (1H, br s, NH), 7.50–7.37 (6H, m, Ar-H), 7.13–7.06 (2H, m, Ar-H), 5.81 (1H, br s, NH), 5.20 (2H, s, CH_2_). δ_C_ (CDCl_3_) 167.1 (C = O), 157.3 (C), 136.8 (C), 133.6 (CH), 132.9 (CH), 129.2 (2 × CH), 128.9 (CH), 128.1 (2 × CH), 121.7 (CH), 121.3 (C), 112.9 (CH), 71.5 (CH_2_). LRMS *m/z* 228.2 (100%, M + H). HPLC purity: 100%. These data are in good agreement with literature values [59].

**2-(Benzyloxy)-*N*-((1-methyl-2-nitro-1*H*-imidazol-5-yl)methyl)benzamide (7).** (1-Methyl-2-nitro-*1H*-imidazol-5-yl)methanamine (**21**). To aldehyde **20** (0.60 g, 3.87 mmol) in MeOH (16 mL) was added ammonium acetate (5.96 g, 77.4 mmol) and the mixture was stirred at 45 °C for 2 h. NaCNBH_4_ (0.17 g, 2.37 mmol) was added and the mixture stirred for 48 h, acidified to pH 1 with 6 M aqueous HCl and solvent removed in vacuo. The aqueous residue was filtered and the filtrate basified to pH 14 with 6 M aqueous KOH. The aqueous was extracted with DCM (3 × 50 mL), then saturated with NaCl and further extracted with DCM (3 × 50 mL). The combined organic fractions were concentrated and filtered through a plug of silica (9:1, DCM, MeOH) to give a crude residue **21** (0.15 g) which was used without further purification. δ_H_ ((CD_3_)_2_SO) 7.07 (1H, s, H-4), 3.90 (3H, s, CH_3_), 3.77 (2H, s, CH_2_), 3.17 (2H, br s, NH_2_). 

To benzoic acid **26** (0.16 g, 0.704 mmol) in DCM (3 mL) was added oxalyl chloride (0.30 mL, 3.54 mmol) and the mixture was stirred overnight at room temperature. Solvent was removed in vacuo. The crude residue was taken up in DCM (2 mL) and added to a solution of amine **21** (0.10 g, 0.64 mmol) in pyridine (2 mL) at 0 °C. The mixture was allowed to come to room temperature and stirred for 4 h. Solvent was removed in vacuo and the crude residue was purified by column chromatography (2:1, EtOAc, X4) to yield **7** (0.16 g, 70%) as yellow solid, mp 120–122 °C. δ_H_ ((CD_3_)_2_SO) 8.62 (1H, t, *J* = 5.5 Hz, NH), 7.67 (1H, dd, *J* = 7.6, 1.7 Hz, H-6), 7.49–7.43 (1H, m, H-4), 7.42–7.38 (2H, m, Ar-H), 7.33–7.26 (3H, m, Ar-H), 7.22 (1H, d, *J* = 7.8 Hz, H-3), 7.07–7.02 (2H, m, H-5, H-4′), 5.19 (2H, s, OCH_2_), 4.55 (2H, d, *J* = 5.6 Hz, NCH_2_), 3.75 (3H, s, NCH_3_). δ_C_ ((CD_3_)_2_SO) 165.5 (C = O), 155.8 (C), 145.4 (C), 136.4 (C), 136.0 (C), 132.1 (CH), 130.0 (CH), 128.3 (2 × CH), 127.9 (CH), 127.6 (2 × CH), 127.2 (CH), 123.8 (C), 120.7 (CH), 113.3 (CH), 70.0 (CH_2_), 33.9 (CH_3_), 33.1 (CH_2_). HRMS calcd for C_19_H_19_N_4_O_4_ (M + H) *m/z* 367.1401, found 367.1407. LRMS *m/z* 367.2 (100%, M + H). HPLC purity: 99.3% (effector **6**: not detected).

**2-(Benzyloxy)-*N*-(1-(1-methyl-2-nitro-1*H*-imidazol-5-yl)ethyl)benzamide (8).** 1-(1-Methyl-2-nitro-*1H*-imidazol-5-yl)ethan-1-amine (**24**). To ketone **23** (0.20 g, 1.2 mmol) in MeOH (4 mL) was added ammonium acetate (1.8 g, 23.6 mmol) and the resulting solution was stirred at 40 °C for 1 h. Activated 4 Å molecular sieves were added and the mixture was stirred a further 1 h. The mixture was cooled to 0 °C and NaCNBH_4_ (0.05 g, 0.84 mmol) added, then the mixture was warmed to room temperature and stirred overnight. Solvent was removed in vacuo and the resulting residue filtered through a plug of silica (9:1, DCM, MeOH) to give a crude residue 24 (0.04 g) which was used without further purification. δ_H_ ((CD_3_)_2_SO) 7.10 (1H, s, H-4), 4.05 (1H, q, *J* = 6.7 Hz, CH), 3.94 (3H, s, NCH_3_), 1.38 (3H, d, *J* = 6.7 Hz, CH_3_).

To benzoic acid **26** (0.06 g, 0.26 mmol) in DCM (1 mL) added oxalyl chloride (0.11 mL, 3.54 mmol) and the mixture was stirred overnight at room temperature. Solvent was removed in vacuo. The crude residue was taken up in DCM (1 mL) and added to a solution of amine **24** (0.04 g, 0.24 mmol) in pyridine (0.5 mL) at 0 °C. The mixture was allowed to come to room temperature and stirred overnight. Solvent was removed in vacuo and the crude residue was purified by column chromatography (2:1, X4, EtOAc) to yield **8** (0.026 g, 29%) as a white solid, mp 151–153 °C. δ_H_ (CDCl_3_) 8.22 (1H, dd, *J* = 7.8, 1.8 Hz, H-6), 8.08 (1H, d, *J* = 8.2 Hz, NH), 7.54–7.50 (1H, m, H-4), 7.43–7.28 (5H, m, Ar-H), 7.15 (1H, td, *J* = 7.2, 1.0 Hz, H-5), 7.10 (1H, d, *J* = 8.3 Hz, H-3), 6.81 (1H, d, *J* = 0.6 Hz, H-4′), 5.36 (1H, p, *J* = 8.0 Hz, CH), 5.08 (2H, ddd, *J* = 10.8, 9.8, 9.8 Hz, OCH_2_), 3.84 (3H, s, NCH_3_), 1.44 (3H, d, *J* = 6.9 Hz, CH_3_). δ_C_ (CDCl_3_) 164.5 (C = O), 157.1 (C), 139.3 (C), 135.0 (C), 133.7 (CH), 132.7 (CH), 129.5 (CH), 129.2 (2 × CH), 128.4 (2 × CH), 126.0 (CH), 122.2 (CH), 120.8 (C), 112.5 (CH), 71.9 (CH_2_), 39.8 (CH), 34.4 (CH_3_), 19.5 (CH_3_); C-NO_2_ was not observed. HRMS calcd for C_20_H_21_N_4_O_4_ (M + H) *m/z* 381.1571, found 381.1568. LRMS *m/z* 381.2 (100%, M + H). HPLC purity: 99.8% (effector **6**: not detected).

**2-((1-Methyl-2-nitro-1*H*-imidazol-5-yl)methyl)phthalazin-1(2*H*)-one (10).** To phthalazinone **9** (0.205 g, 1.40 mmol) in DMF (2 mL) was added NaH (60% dispersion in mineral oil) (0.084 g, 2.10 mmol) and the resulting solution stirred for 30 min at room temperature. Nitroimidazole **19** (0.27 g, 1.54 mmol) was added and the solution was stirred for 30 min, then quenched on ice. The resulting suspension was filtered, the collected solid washed with water (5 mL), X4 (5 mL) and dried in vacuo to yield **10** (0.04 g, 10%) as a yellow solid, mp 207–209 °C. δ_H_ ((CD_3_)_2_SO) 8.49 (1H, s, H-4), 8.29 (1H, dd, *J* = 7.84, 0.7 Hz, H-8), 7.98 (2H, d, *J* = 3.7 Hz, H-7, H-6), 7.93–7.87 (1H, m, H-5), 7.19 (1H, s, H-4′), 5.47 (2H, s, CH_2_), 4.00 (3H, s, CH_3_). δ_C_ ((CD_3_)_2_SO) 158.4 (C = O), 145.6 (C), 138.9 (C), 134.1 (C), 133.9 (C), 132.4 (CH), 129.3 (C), 128.4 (CH), 127.1 (CH), 127.9 (C), 125.9 (CH), 43.6 (CH_2_), 34.4(CH_3_). HRMS calcd for C_13_H_12_N_5_O_3_ (M + H) *m/z* 286.0935, found 286.0941. LRMS *m/z* 286.1 (100%, M + H). HPLC purity: 94.4% (effector **10**: 0.1%).

**2-((1-Methyl-2-nitro-1*H*-imidazol-5-yl)methyl)isoquinolin-1(2*H*)-one (11).** To isoquinolinone **2a** (0.10 g, 0.69 mmol) in DMF (2 mL) was added a 1.0 M THF solution of LiHMDS (0.90 mL, 0.90 mmol) and the mixture was stirred for 2 h. Nitroimidazole **19** (0.12 g, 0.69 mmol) in DMF (2 mL) was added dropwise and the reaction was stirred overnight. Water (2 mL) was added and the resulting mixture extracted with EtOAc (2 × 5 mL). The organic fractions were washed with water (5 mL), brine (5 mL) and dried over Na_2_SO_4_. Solvent was removed in vacuo and the crude product was purified by column chromatography (2:1, DCM, EtOAc) to yield **11** (0.11 g, 55%) as a yellow solid, mp 207–210 °C. δ_H_ ((CD_3_)_2_SO) 8.25 (1H, dt, *J* = 8.1, 0.6 Hz, H-8), 7.74 (1H, ddd, *J* = 9.2, 7.0, 1.3 Hz, H-6), 7.68 (1H, dd, *J* = 7.9, 0.7 Hz, H-7) 7.55 (2H, d, *J* = 7.6 Hz, H-5, H-3), 7.16 (1H, s, H-4′), 6.71 (1H, d, *J* = 7.4 Hz, H-4), 5.33 (2H, s, CH_2_), 3.99 (3H, s, CH_3_). δ_C_ ((CD_3_)_2_SO) 161.0 (C = O), 145.7 (C), 136.9 (C), 134.6 (C), 132.7 (CH), 132.0 (CH), 127.8 (CH), 127.1 (CH), 127.0 (CH), 126.3 (CH), 125.2 (C), 106.0 (CH), 40.9 (CH_2_), 34.3 (CH_3_). HRMS calcd for C_14_H_13_N_4_O_3_ (M + H*) m/z* 285.0982, found 285.0978. LRMS 2 *m/z* 85.1 (100%, M + H). HPLC purity: 96.7% (effector **2a**: not detected).

**8-Hydroxy-2-methylquinazolin-4(3*H*)-one (12).** To a stirred solution of quinazolinone **17** (400 mg, 2.10 mmol) in dry DCM (10 mL) was added a 1.0 M DCM solution of BBr_3_ (4.8 mL, 4.8 mmol) and the mixture stirred at 30 °C for 72 h. Solvent was removed in vacuo, the residue cooled to 5 °C and aqueous NaOH (2.5 M, 30 mL) was added. The mixture was stirred at 20 °C for 15 min, filtered and the pH of the filtrate adjusted to 5 with aqueous HCl (6 M). The mixture was chilled at 5 °C for 1 h. The resulting precipitate was purified by column chromatography (1:1–1:0, EtOAc, X4) to yield **12** (0.23 g, 63%) as a tan powder, mp 261–263 °C (lit. [47] mp 253–258 °C). δ_H_ ((CD_3_)_2_SO) 12.14 (1H, br s, NH), 9.36 (1H, s, OH), 7.49 (1H, dd, *J* = 7.9, 1.4 Hz, H-5), 7.25 (1H, t, *J* = 7.9 Hz, H-6), 7.14 (1H, dd, *J* = 7.8, 1.4 Hz, H-7), 2.37 (3H, s, 2-CH_3_). δ_C_ ((CD_3_)_2_SO) 162.2 (C = O), 153.0 (C), 152.7 (C), 138.4 (C), 126.7 (CH), 121.9 (C), 118.6 (CH), 115.9 (CH), 21.9 (CH_3_). LRMS *m/z* 177.2 (100%, M + H). These data are in good agreement with literature values [47,60].

**2-Methyl-8-((1-methyl-2-nitro-*1H*-imidazol-5-yl)methoxy)quinazolin-4(*3H*)-one (13).** To phenol **12** (0.069 g, 0.39 mmol) in DMF (2 mL) was added K_2_CO_3_ (0.16 g, 1.16 mmol) followed by nitroimidazole **19** (0.069 g, 0.57 mmol). The mixture was stirred for 72 h, diluted with water and collected by filtration. The crude compound was purified by column chromatography (40:1, DCM, MeOH) to yield **13** (0.057 g, 47%) as a white solid, mp 253– 256 °C. δ_H_ ((CD_3_)_2_SO) 12.24 (1H, br s, NH), 7.72 (1H, dd, *J* = 8.0, 1.1 Hz, H-5), 7.54 (1H, dd, *J* = 8.0, 1.1 Hz, H-7), 7.39 (1H, t, *J* = 8.0 Hz, H-6), 7.39 (1H, s, H-4′), 5.38 (2H, s, CH_2_), 4.06 (3H, s, NCH_3_), 2.34 (3H, s, CH_3_). δ_C_ ((CD_3_)_2_SO) 161.5 (C = O), 153.4 (C), 152.1 (C), 140.5 (C), 133.5 (C), 128.6 (CH), 125.9 (CH), 122.0 (C), 119.2 (CH), 118.7 (CH), 61.0 (CH_2_), 34.3 (CH_3_), 21.6 (CH_3_); C-NO_2_ was not observed. HRMS calcd for C_14_H_14_N_5_O_4_ (M + H) *m/z* 316.1040, found 316.1039. LRMS *m/z* 316.1 (100%, M + H). HPLC purity: 95.6% (effector **12**: 0.4%).

**8-Methoxy-2-methylquinazolin-4(3*H*)-one (17).** A mixture of acid **16** (8.37 g, 42.5 mmol) and Pd/C (5%, 0.10 g) in EtOH (100 mL) was stirred under H_2_ (60 psi) for 16 h. The mixture was filtered through diatomaceous earth, the pad washed with EtOH (50 mL) and solvent removed in vacuo to give 2-amino-3-methoxybenzoic acid (7.05 g, 99%) as a tan powder which was used directly. CDI (6.70 g, 41.3 mmol) was added to a stirred solution of 2-amino-3-methoxybenzoic acid (7.05 g, 42.2 mmol) in dry DMF (100 mL) at 70 °C and the mixture stirred for 10 min. A solution of aqueous NH_3_ (108 mL, 1.05 mol) was added slowly and the mixture was stirred at 70 °C for 16 h. The mixture was cooled to 20 °C, diluted with EtOAc (300 mL), washed with water (3 × 100 mL), brine (100 mL), dried (MgSO_4_) and solvent removed in vacuo to give crude 2-amino-3-methoxybenzamide (5.21 g, 74%) as a tan solid which was used directly. Acetyl chloride (1.43 mL, 20.2 mmol) in dry THF (2 mL) was added dropwise to a stirred solution of 2-amino-3-methoxybenzamide (1.66 g, 10.0 mmol) and pyridine (1.06 mL, 13.0 mmol) in dry THF (50 mL) at 20 °C and the resulting mixture stirred for 16 h. Solvent was removed in vacuo, the residue dissolved in aqueous NaOH (0.5 M, 25 mL) and stirred for 1 h. The solution was carefully neutralized with aqueous HCl (0.5 M) and stirred at 5 °C for 1 h. The precipitate was filtered and recrystallized from MeOH/water to yield **17** (0.91 g, 48%) as a white solid, mp 263–266 °C (lit. [61] mp 262–263 °C). δ_H_ ((CD_3_)_2_SO) 10.79 (1H, br s, NH), 7.61 (1H, dd, *J* = 7.8, 1.4 Hz, H-5), 7.36 (1H, t, *J* = 7.9 Hz, H-6), 7.30 (1H, dd, *J* = 8.0, 1.4 Hz, H-7), 3.87 (3H, s, OCH_3_), 2.34 (3H, s, CH_3_). δ_C_ ((CD_3_)_2_SO) 161.7 (C = O), 154.0 (C), 152.9 (C), 139.4 (C), 126.0 (CH), 121.6 (C), 116.7 (CH), 114.7 (CH), 55.8 (CH_3_), 21.5 (CH_3_). LRMS *m/z* 191.1 (100%, M + H). These data are in good agreement with literature values [60,61].

**5-(Chloromethyl)-1-methyl-2-nitro-1*H*-imidazole (19).** To a solution of alcohol **18** (0.50 g, 3.2 mmol) in THF (10 mL) at 0 °C was added diisopropylethylamine (0.67 mL, 3.8 mmol) and methanesulfonyl chloride (0.3 mL, 0.38 mmol) and the mixture was stirred for 30 min. EtOAc (20 mL) was added and the solution washed with 1M aqueous HCl (20 mL), dried over MgSO_4_ and solvent removed in vacuo. The crude material was purified by column chromatography (2:1, X4, EtOAc) to yield **19** (0.51 g, 91%) as a yellow solid, mp 99–100 °C (lit. [43] mp 94–96 °C). δ_H_ (CDCl_3_) 7.19 (1H, s, 4-H), 4.63 (2H, s, CH_2_), 4.08 (3H, s, CH_3_). δ_C_ (CDCl_3_) 146.4 (C-NO_2_), 132.9 (C), 128.6 (CH), 34.3 (CH_3_), 33.9 (CH_2_). LRMS *m/z* 176.1 (100%, M^35^ + H), 178.1 (36%, M^37^ + H). These data are in good agreement with literature values [43].

**1-Methyl-2-nitro-*1H*-imidazole-5-carbaldehyde (20).** To alcohol **18** (1.0 g, 6.36 mmol) in chloroform (70 mL) was added MnO_2_ (2.76 g, 31.8 mmol) and the mixture was heated to reflux overnight. After cooling, the resulting slurry was filtered through diatomaceous earth and solvent removed in vacuo to yield **19** (1.5 g, 76%) as a pale yellow solid, mp 112–114 °C (lit. [62] mp 114–115 °C). δ_H_ (CDCl_3_) 9.93 (1H, s, CHO), 7.81 (1H, s, H-4), 4.36 (3H, s, CH_3_). δ_C_ (CDCl_3_) 180.4 (C = O), 148.3 (C-NO_2_), 139.4 (CH), 132.4 (CH), 35.6 (CH_3_). LRMS 188.1 (100%, M + CH_3_OH). These data are in good agreement with literature values [35].

**1-(1-Methyl-2-nitro-*1H*-imidazol-5-yl)ethan-1-ol (22).** A 3 M solution of MeMgBr in Et_2_O (0.32 mL, 0.97 mmoL) was added to a solution of TiCl_4_ (0.11 mL, 0.97 mmol) in DCM (4 mL) at –78 °C. The mixture was allowed to warm to –30 °C and added to a solution of aldehyde **20** (0.15 g, 0.97 mmol) in DCM (4 mL) at –30 °C. This mixture was maintained at –30 to –20 °C for 3 h, then quenched by addition of saturated aqueous NH_4_Cl (5 mL). The aqueous fraction was extracted with DCM (3 × 10 mL) and solvent removed in vacuo. The crude mixture was purified by column chromatography (50:1, DCM, MeOH) to yield **22** (0.09 g, 53%) as an orange solid, mp 101–103 °C. δ_H_ (Acetone*-d6*) 7.06 (1H, s, H-4), 5.05–4.94 (1H, m, CH), 4.59 (1H, br s, OH), 4.08 (3H, s, NCH_3_), 1.61 (3H, d, *J* = 6.6 Hz, CH_3_). δ_C_ (Acetone*-d6*) 147.3 (C-NO_2_), 141.6 (C), 124.7 (CH), 60.3 (CH), 33.9 (CH_3_), 21.1 (CH_3_). LRMS *m/z* 172.2 (100%, M + H). These data are in good agreement with literature values [49].

**1-(1-Methyl-2-nitro-*1H*-imidazol-5-yl)ethan-1-one (23).** To a solution of alcohol **22** (0.06 g, 0.35 mmol) in CHCl_3_ (10 mL) was added MnO_2_ (0.20 g, 2.3 mmol) and the resulting solution was stirred at reflux overnight. The resulting slurry was filtered through diatomaceous earth and solvent was removed in vacuo to yield **23** (0.048 g, 81%) as a yellow solid, mp 71–72 °C (lit. [62] mp 81–83 °C). δ_H_ (CDCl_3_) 7.77 (1H, s, H-4), 4.32 (3H, s, NCH_3_), 2.59 (3H, s, CH_3_). δ_C_ (CDCl_3_) 189.2 (C = O), 148.2 (C-NO_2_), 135.7 (CH), 132.3 (C), 35.8 (CH_3_), 28.6 (CH_3_). HRMS calcd for C_6_H_7_N_3_O_3_ (M + H) *m/z* 170.0560, found 170.0556. LRMS *m/z* 170.1 (100%, M + H).

**2-(Benzyloxy)benzoic acid (26).** To ester **25** (1.60 mL, 12.04 mmol) in MeCN (50 mL) was added K_2_CO_3_ (3.30 g, 24.08 mmol) followed by benzyl bromide (1.50 mL, 12.88 mmol) and the resulting mixture was stirred at reflux overnight. After cooling the reaction was diluted with water (25 mL) and the aqueous fraction extracted with DCM (3 × 20 mL). The combined organic extracts were washed with water (25 mL), dried over Na_2_SO_4_ and solvent removed in vacuo. The crude residue was dissolved in 1,4-dioxane (50 mL) and aqueous NaOH (1.5 M, 12 mL) was added. The solution was stirred at reflux for 2 h, cooled to room temperature and the organic residues removed in vacuo. The remaining solution was acidified with 1M aqueous HCl, extracted with EtOAc (3 × 25 mL), the organic extracts dried over Na_2_SO_4_ and solvent removed in vacuo to yield **26** (2.4 g, 82%) as a colourless solid, mp 70–71 °C (lit. [63] mp 73–75 °C). δ_H_ (CDCl_3_) 10.72 (1H, br s, OH), 8.21 (1H, dd, *J* = 7.8, 1.8 Hz, H-6), 7.56 (1H, ddd, *J* = 9.2, 7.4, 1.8 Hz, H-4), 7.46–7.38 (5H, m, Ar-H), 7.18–7.11 (2H, m, H-5, H-3) 5.30 (2H, s, CH_2_). δ_C_ (CDCl_3_) 165.6 (C = O), 157.6 (C), 135.2 (CH), 134.5 (C), 134.0 (CH), 129.34 (CH), 129.32 (2 × CH), 128.1 (2 × CH), 122.6 (CH), 118.2 (C), 113.3 (CH), 72.4 (CH_2_). LRMS *m/z* 229.2 (100%, M + H). These data are in good agreement with literature values [64].

### 3.2. Assays

#### 3.2.1. Radiolytic Reduction

DMSO stock solutions (10 mM) were transferred to a Pd/C catalyzed anaerobic chamber with an atmosphere of 5% H_2_, 5% CO_2_, 90% N_2_ (Shellab Bactron, Sheldon Manufacturing Inc. Cornelius, OR), and diluted to make 10 μM solutions in 100 mM sodium formate/5 mM sodium phosphate, pH 7.0, using buffer that had been equilibrated in the chamber for ≥3 days to ensure deoxygenation. Aliquots (1 mL) were then transferred to HPLC vials, sealed, removed from the chamber and irradiated using an Eldorado 78 teletherapy cobalt-60 unit. The dose rate was ~3 Gy/min, with exact values determined by Fricke dosimetry [65]. Samples were analyzed immediately by LC/MS, or after incubating for 5 h at 37 °C. 

#### 3.2.2. Chemical Reduction

Compound **3c** (2.14 mg, 5.9 µmol), nitrogen purged acetonitrile, ammonium chloride, and zinc powder were transferred to a Pd/C catalyzed anaerobic chamber with an atmosphere of 5% H_2_, 5% CO_2_, 90% N_2_ (Shellab Bactron, Sheldon Manufacturing Inc. Cornelius, OR). **3c** was dissolved in acetonitrile (1 mL), followed by addition of NH_4_Cl (32.1 mg, 0.60 mmol) and zinc powder (32.6 mg, 0.50 mmol). The resulting slurry was mixed and left for 1 h, then filtered through 0.2 µm acrodisc filters. Filtrate samples were diluted in mobile phase (1:100) and analyzed immediately by LC/MS, or after 3 h incubation at room temperature.

#### 3.2.3. LC/MS Analysis

Solutions were analyzed with an Agilent 1260 series HPLC coupled with an Agilent single stage quadrupole mass spectrometer, equipped with a Jet Stream^TM^ electrospray ionization source (Agilent Technologies, USA). Chromatographic separation was achieved on an Agilent Zorbax C18, 3.0 × 150 mm column with 5 μm particle size, maintained at 30 °C. The mobile phase consisted of 45 mM ammonium formate buffer, pH 3.5 (A) and 80% acetonitrile with 0.01% formic acid (B) with a linear gradients (40–100% B over 8 min, 100–40% B over 1 min, held at 40% B for 3 min) at a flow rate of 0.5 mL/min. The sample volume injected was 100 µL and the auto sampler was set at 4 °C. The mass spectrometer was run in positive and negative ion ionization modes, with dual polarity scans from 100–600 m/z. Instrument parameters of the mass spectrometer were: fragmentation voltage 70 V, gas flow 12 L/min, gas temperature 250 °C, sheath gas flow 10 L/min, sheath gas temperature 325 °C, nebuliser 35 psi, capillary voltage 2500 V, and nozzle voltage 500V. Absorbance detection was at 276 nm (bandwidth 16 nm) with quantitation by integration of peak areas with Agilent Open Lab CDS Chemstation software. Unirradiated samples were used for single-point calibration. 

#### 3.2.4. Biochemical PARP Inhibition Assay

Inhibition of PARP-1 activity was evaluated in a radioisotope-based filter binding assay by Reaction Biology Corporation (Malvern PA) using human recombinant PARP-1 (residues 1-1014, Genbank accession # NM_001618.3, MW = 114.8 kDa) expressed with a C-terminal His-tag in *Sf9* insect cells, >80% pure by SDS-PAGE. Polymerization of ^32^P-NAD^+^ (10 µM) on core histones (0.01 mg/mL) in buffer (50 mM Tris-HCl, 50 mM NaCl, 10 mM MgCl_2_, 0.02% Brij35, 1 mM DTT, 1% DMSO and 20 µg/mL activated DNA (Sigma-Aldrich D4522) was measured after 1 h incubation and washout of remaining free NAD^+^. Each compound was included in a 10-point, 3-fold dilution dose response experiment and an IC_50_ value derived.

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
