# Peer review of "Studies Towards Hypoxia-Activated Prodrugs of PARP Inhibitors"

_molecules, 2019, doi:10.3390/molecules24081559_

Round 1

Reviewer 1 Report

This manuscript reports studies on the development of reductively activated inhibitors of PARPs. This is not a new concept but these studies do add significantly to previous work and the manuscript is worthy of publication, subject to revision as outlined below. The manuscript is written clearly in good scientific English and many of the compounds are new. The compounds are well characterised.

Some of the percentage yields are given to a precision of 1 decimal place; this is inappropriate and these values should be rounded to the nearest integer %.

IC50 values are presented in Table 1 for only two pairs of drug/prodrug. Since the aim of the project was to identify pairs where the inhibitory activities were markedly different (a requirement for a prodrug system), surely all pairs of drug/prodrug should have been tested. I recommend that all the remaining pairs be evaluated for inhibition of PARP-1 and the IC50 values should be given in the revised manuscript. Moreover, no SD values (±) are given for the data in this table, so it is difficult to judge the significance of the differentials. All IC50 values should be accompanied by the pIC50 values and the ± values for the pIC50s.

The discussion on page 9 (lines 200-214) explains the difference in release of the PARPi between chemically-reduced reductions reported earlier and the radiolytic method reported here and ascribes the release of PARPi in the earlier studies to complexation with metals facilitating fragmentation in the chemical studies. This is unlikely in the case of the NaBH4/Pd-C system, where the metal in not in solution. An alternative explanation might be that the radiolytic method gives the corresponding hydroxylamine by a 4e reduction (as demonstrated in this manuscript), whereas the chemical reduction systems give the corresponding amine by a 6e reduction. It could be that the hydroxylamines are insufficiently electron-donating to drive the fragmentation. Furthermore, it is known in other bioreductive systems (e.g. indole­quinones), that the rate of fragmentation depends markedly on the leaving-group ability of the delivered “drug”.1

The conclusion (page 10, lines 35-40) points towards O-linked nitroimidazolylmethyl prodrugs as releasing the PARPi effectively, although the difference in inhibitory activity is modest as the key H-bonds are not masked as the authors comment. 1-(Nitrothiophenylmethyloxy)isoquinolines are synthetically accessible (as are their indolequinone analogues) and could make good O-linked bioreductively activated prodrugs in which both the isoquinolinone N-H and the carbonyl=O are masked in the prodrug.2

The authors should discuss the evidence that alkylation of the phthalazinones and of isoquinolinones has taken place at nitrogen, rather than at oxygen, in the light of the reported capricious nature of alkylations of isoquinolinones.2 The 13C NMR data suggest that alkylation has occurred at N, as shown in the manuscript, but this should be discussed briefly.

Page 13, line 356: The 1H NMR signal for the OCH2 protons is cited as q, J = 10.2 Hz. This cannot be correct, as there are no neighbouring protons to which they could couple by such a coupling constant. As there is a chiral centre in the molecule, these OCH2 protons will be diastereotopic and magnetically inequivalent. Thus they should give two separate doublet signals, each d, 2J = 10.2 Hz.

Much more detail should be given for the PARP-1 inhibition assays, including the source and purity of the enzyme, as slight differences in method are known to give different IC50 values for PARPi.

Page 7, lines 150,151: Amine 24 is being acylated, not alkylated.

Page 7, lines 158: 14/15 should be 12/13.

There are several errors of punctuation in the manuscript, particularly missing hyphens.

References

1.         Swann, E.; Moody, C. J.; Stratford, M. R. L.; Patel, K. B.; Naylor, M. A.; Vojnovic, B.; Wardman, P.; Everett, S. A. Rates of reductive elimination of substituted nitrophenols from the (indol-3-yl)methyl position of indolequinones. J. Chem. Soc., Perkin Trans. 2 2001, 1340-1345.

2.         Ferrer, S.; Naughton, D. P.; Parveen, I.; Threadgill, M. D. N- and O-Alkylation of isoquinolin-1-ones in the Mitsunobu reaction: Development of potential drug delivery systems. J. Chem. Soc., Perkin Trans. 1 2002, 335-340.

Author Response

Response in attached PDF.

Reviewer 2 Report

Manuscript  ID molecules-466988 entitled "Studies towards hypoxia-activated prodrugs of PARP inhibitors" by Dickson B.D. et al describe the evaluation of a set of 2-nitroimidazolyl derivatives as potential prodrug of PARP inhibitors. Autors, after having synthesized both PARPi and the corresponding prodrugs, assayed selected compounds against PARP-1 and then evaluated if potential prodrugs 3c,5,7,8,10,11,13 were able to release the relevant PARPi. In conclusion Authors found out that only 13 release the PARPi 12.

I found this manuscript very clear and well presented, but the following points need to be revised.

Introduction 

-pag 2, lines 50-51:   when referring to the PARP involvement in the wider DNA damage cellular response, include reference "Gallorini, M.; Maccallini, C.; Ammazzalorso, A.; Amoia, P.; De Filippis, B.; Fantacuzzi, M.; Giampietro, L.; Cataldi, A.; Amoroso, R. The Selective Acetamidine-Based iNOS Inhibitor CM544 Reduces Glioma Cell Proliferation by Enhancing PARP-1 Cleavage In Vitro. Int. J. Mol. Sci. 2019, 20, 495." , as it is a recent example of the  crucial role of PARP  in the antiglioma effects of nitric oxide inhibition.

pag. 4 Figure 3: below each chemical structure, the molecule number should be reported at first, therefore, for example, 4: R=H; 5: R= A, and so on.

Results and discussion

pag.7 lines 157-158. Explain why only compounds couples 4/5 and 12/13 where assayed against PARP-1. Inhibition values should be presented for all PARPi at least.

pag.8, Table 2. For compound 13, explain why % of PARPi formed is greater than % of prodrug loss. 

pag 9, lines 205-206: the 3c hydroxylamine was recognized as the main product of reduction, did the  authors test if it is inactive as PARP-1 inhibitor?

Materials and Methods

The CNMR spectra of some compounds is not reported (4, 6, 12, 17,19, etc), please include them.

Author Response

Response in attached PDF.

Reviewer 3 Report

In Medicinal Chemistry phthalazinones provide derivatives able to interact with different kinds of biological targets.

Poly-[ADP-ribose] polymerase (PARP) -1 inhibitory activity detected for some phthalazine derivatives points to the  discovery of a number of potent PARP-1 antagonists that bear the phthalazinone core (for instance olaparib) .

The authors propose a  useful prodrug strategy and  investigate potential prodrug approaches for PARP  inhibitors,  to limit exposure of normal cells to  active drug, improving their  tumour selectivity .    Preparation of model compounds is described and discussed  in detail . The findings add new knowledge on how to handle PARP inhibitors and address their effect specifically to tumor targets.

The article is well organized and well written.

Some references on PARP are rather old, an could be updated.

Minor point

The article is acceptable after updating the references indicated above.

Author Response

Response in attached PDF.

Round 2

Reviewer 2 Report

Response 1: We thank the reviewer for pointing out this interesting manuscript that demonstrates PARP-1 cleavage in response to an iNOS inhibitor in a glioma cell line. However, PARP-1 cleavage in this context is a marker of apoptosis, not an indicator of PARP-1 inhibition and is not necessarily related to DNA damage responses. Thus we can’t see the relevance of this paper to our manuscript.

This reference was suggested since it is a recent study supporting experimental evidences about the central role that PARP direct or indirect inhibition (or inactivation) has in DNA repair. In this study, in fact, iNOS inhibition mediated PARP cleavage which induces glioma cells necrosis, with cell cycle arrest and antiproliferative effects. Moreover this work supports the involvement of PARP-1 in glioma chemoresistance, confirming Authors statement that “Due to this wide ranging involvement, PARPi can sensitise cells to a variety of DNA  damaging agents and so combination with cytotoxic chemotherapies or radiotherapy has been  proposed as an approach for treatment of HRR competent tumours [19,20].”

Therefore I confirm that this study supports that PARPi could be useful antitumoral agents and that it should be cited.

Response 3: The reviewer noted that IC50 values were only provided for two pairs of compounds and requested that we provided IC50 values for all pairs of drug and prodrug. Our main aim in this project was assessment of 2-nitroimidazolyl triggers attached to the amide of scaffolds related to PARP inhibitors and their fragmentation. The majority of these scaffolds have previously been reported in the literature and are known to be poor PARP inhibitors (IC50 values >1 µM1 ), however they form the core of more developed PARP inhibitors2 . Further it has been demonstrated that simple methylation of the amide nitrogen of a PARP inhibitor is sufficient to abrogate PARP inhibition as exemplified by Griffin et al Table 1 compound 6 vs. compound 433 . Consequently we considered this additional data would not provide any further meaningful information and that assessment of a representative compound from the two prodrug approaches (5 and 13) would be sufficient to confirm abrogation of PARP inhibition as shown previously in the literature4 . We acknowledge that our aims in this project were not necessarily clear in the introduction of the manuscript and as such we have added clarification (lines 20, 97 – 98) which we hope indicates clearly that these compounds are models of PARP inhibitors and related HAPs and designed to investigate fragmentation.

It is quite clear that prodrugs are poor active as PARPi. My request was referred to molecules 2a, 2c, 4, 6, 9 and 12
